# Eye-tracking measures of oculomotor speed and control as markers of cognitive ability in Malawian adolescent population: Secondary analysis of a randomized controlled trial

Karoliina Videman[1,2]*, Ulla Ashorn[1], Per Ashorn[1,2], Lotta Hallamaa[1], Kenneth Maleta[3], Charles Mangani[3], Jukka M. Leppänen[4]

1 Center for Child, Adolescent and Maternal Health Research, Tampere University, Faculty of Medicine and Health Technology, Tampere, Finland, 2 Department of Paediatrics, Tampere University Hospital, Tampere, Finland, 3 School of Global and Public Health, Kamuzu University of Health Sciences, Blantyre, Malawi, 4 Department of Psychology and Speech-Language Pathology, University of Turku, Turku, Finland

* karoliina.videman@tuni.fi

## Abstract

Processing speed and response control are fundamental properties of brain function and potential markers of cognitive ability. This study, a secondary analysis of a randomized controlled trial, examined whether eye-tracking measures of saccadic reaction time and gaze control are associated with an established cognitive ability test, Raven's coloured progressive matrices (CPM), among 13-year-old rural Malawian adolescents (1003 participants, 50.3% boys). Mean prosaccadic reaction time ($_pSRT_m$), antisaccade error rate (PE) and CPM result were obtained for 760 (75.8%), 621 (61.6%) and 997 (99.4%) children. Pearson correlation and linear regression were used to evaluate the association of the tasks. Faster $_pSRT_m$ and lower PE were very weakly associated with higher CPM score (rs -0.12, p = .001 and -0.11, p = .006). In the covariate adjusted regression models, faster prosaccadic reaction time (pSRTm) was very weakly associated with higher scores in CPM test (adjusted coef -0.02, 95%CI (-.03- -.002), p = .03), but antisaccadic errors were not associated with CPM score (adjusted coef -0.63, 95%CI (-1.60 -.33), p = .20). Post hoc-analyses suggested stronger associations between eye-tracking measures and CPM among participants with more schooling (years in school <4.5 or >4.5, rs between $_pSRT_m$ and CPM -0.05 and -0.21; between PE and CPM -0.01 and -0.39). The results confirm the predicted association between saccadic speed and cognitive ability in an understudied population, but the connection is weaker than expected according to earlier studies. Schooling potentially moderates the association between eye-tracking tests and CPM.

**Data availability statement:** Data contain potentially identifying and sensitive participant information and cannot be uploaded in an open repository or shared as a supplemental file. Data are available upon request. Data requests can be made to dean John Phuka (ed-sogaph@kuhes.ac.mw), Kamuzu university of Health sciences. Long term data storage and availability will be assured by the study group own repository.

**Funding:** This study was funded by personal grants (received by KV) from the Foundation for Pediatric Research in Finland; State funding for university-level health research, Tampere University Hospital, Wellbeing services county of Pirkanmaa under project numbers 9AB082 and 9AC097 (both received by KV); The Finnish Medical Foundation (received by KV); the Arvo and Lea Ylppö Foundation (received by KV); and Märta Donner grant from Finnish Pediatric Neurology Society (received by KV). The original LAIS study was supported by grants from the grants from the Academy of Finland (grants 79787 and 207010), Foundation for Pediatric Research in Finland and the Medical Research Fund of Tampere University Hospital (all received by PA). The funders had no role in study design, data collection and analysis, decision to publish, or preparation of the manuscript.

**Competing interests:** The authors have declared that no competing interests exist.

## Introduction

Childhood developmental assessment is becoming more important as global emphasis is expanding from reducing child mortality to improving health and developmental trajectories [1,2]. Sustainable development goal (SDG) 4.2 has placed early child development on the global policy agenda, with an emphasis on quality education and lifelong learning opportunities for all [3]. This is an important goal as an estimated 250 million under-5-year-old children are not reaching their developmental potential partly due to malnutrition and other adverse experiences that are common in disadvantaged environments [4]. Loss of developmental potential, without correction, can lead to compromised adolescent and adult cognitive ability [5].

Cognitive development is a hierarchical process in which early brain development and the acquisition of basic information processing capacities (e.g., efficient sampling of visual cues from the environment) lays a foundation for the development of more complex traits and skills (e.g., reasoning, reading) [6]. This process of formation is fast during the first few years of life and continues through adolescence until early adulthood [7]. It has been proposed that assessment of basic brain functions, such as the speed of sensory processing and sensory-motor decisions, may help in monitoring neurocognitive changes over the course of early childhood development [8] or neurocognitive disease progression [9].

Eye movements are among the few behaviors that can be assessed objectively across age groups, including adolescents, and sociocultural settings [10–12]. Different eye movement tasks have been designed for assessing the speed of sensory-motor response (prosaccades, assessed as the mean latency of eye movements to the onset a new target) and voluntary control of sensory-motor response (i.e., "anti-saccades", assessed as the ability to generate saccades to the opposite direction of a visual target) [13]. These processes have well-mapped neural bases in the frontal eye fields, basal ganglia, and superior colliculus [13,14], improve with age [10,11,15], are predictive of working memory and general intelligence [12,16], and are sensitive to clinical conditions, such as Attention Deficit Hyperactivity Disorder (ADHD) [13], global developmental delay [17], and fetal alcohol syndrome [18]. Hence, measures of saccadic speed and saccade control could have utility as objective measures of neurocognitive development in children in different environments [9].

While most eye movement studies have been conducted in controlled laboratory settings in high-income countries, the use of this measure has been recently expanded to less controlled settings that resemble typical screen viewing situations, to less co-operative participant groups (e.g., infants), and to resource-limited settings common in low- and middle-income countries (LMICs) [19–24]. These studies have shown that assessments of basic cognitive operations such as processing speed, sequence learning, face looking, and recognition memory is feasible in children in low-resource settings [19,23,24]. However, these studies have also shown that the metric properties of current measures, at least in young children, are not sufficient for obtaining reliable data on individual differences in cognition [22,25]. Evidence validating these measures against traditional tests of cognitive capacity in LMICs is also lacking or mixed [20,26].



In the present study with 13-year-old adolescents in rural Malawian low-resource setting, our goal was to examine if the eye-tracking measures of processing speed and control are associated with non-verbal inductive reasoning, which is an established correlate of processing speed and a proxy measure for general cognitive ability [27]. More precisely, we hypothesized that faster eye-tracking measured mean pro-saccadic reaction time ($_pSRT_m$) is associated with better performance in Raven's coloured progressive matrices test (CPM) [28,29]. We also tested a hypothesis that $_pSRT_m$ and antisaccade performance reflect interrelated (i.e., correlated), but partially distinct cognitive processes and therefore, $_pSRT_m$ and antisaccadic percentage error (PE) together have stronger association with CPM score than prosaccadic task result alone. Finally, as environmental factors are known to be important determinants of cognitive development in low-SES environments [30], we also examined whether eye tracking measures had independent contributions to cognitive ability when the impact of SES, maternal and child schooling, child sex and growth were controlled.

## Methods

### Participants

Children of the current study were the offspring of Lungwena antenatal intervention study (LAIS) participants, a clinical trial originally designed to study the effects of different antenatal maternal infection control on the mother and the offspring in rural Malawi [31]. Of the 3358 pregnant women who were approached, 1320 (39.3%) were enrolled to the study and randomly assigned to three study groups: women in the control group received standard Malawian antenatal care, which at the time of the study included intermittent preventive treatment against malaria during pregnancy (IPTp) with sulfadoxine-pyrimethamine (SP, 3 tablets orally, each containing 500 mg of sulfadoxine and 25 mg of pyrimethamine) administered twice during pregnancy and placebo in lieu of azithromycin (AZI). Women in the monthly SP group received SP monthly from enrollment until 37 weeks' gestation and a placebo in lieu of AZI, and women in the AZI-SP group received monthly SP and active azithromycin twice (2 tablets orally, each containing 500 mg of azithromycin): once at enrollment and once between 28 and 34 weeks of gestation. The target population comprised of pregnant women who came to the antenatal care between December 2003 and October 2006. There were 1269 babies born alive (7 sets of twins). During the early follow-up of the offspring, we did not collect information of thyroidal function or bilirubin levels. The detailed study design, including for example inclusion and exclusion criteria and sample size calculation, can be found in the earlier publications [32,33].

The main findings of the original trial were that the prevalence of preterm delivery and low birth weight were lower and mean infant size at 1 month was bigger with intensive infection treatment during pregnancy [31]. A 5-year follow-up for the offspring indicated that the intervention had a positive effect on child development measured with Griffith's mental developmental scales, but there was no difference in cognitive results (CPM) at approximately 13 years of age [33,34]. The intervention reduced the cumulative incidence of stunting until early adolescence, 13 years of age [33]. In this article the data from randomized controlled trial is used to secondary analysis.

### Follow-up at 13 years

The current follow-up was planned as a separate wave of assessment and implemented between January 19, 2018, and March 19, 2019. The study participants were mean 12.8 years old at the time of the follow-up (age range from 10.9 years to 14.6 years). The study team made visits to the homes of children who had participated in the original LAIS trial and were not known to have died. If the participants had moved, the study team attempted to find information on their new location from nearby households. A second visit was arranged at a study clinic to perform growth evaluation, interview, and developmental assessments.

We included all the children whom we could trace in statistical analyses, except those with missing data for a specific outcome. We accounted for age-variation by using age at assessment as a covariate in the analyses.

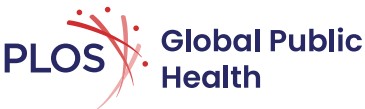

## Ethical statement

Each participant and at least one caregiver provided a new, written informed consent before participation which they signed or thumb printed. The protocol was approved by the College of Medicine Research and Ethics Committee, Malawi, and the Ethical Committee of Pirkanmaa Hospital District, Finland. The trial was registered at www.clinicaltrials.gov (identifier NCT00131235). Both the original trial and the follow-up were performed according to Good Clinical Practice and the ethical standards of the Declaration of Helsinki.

## Outcomes

**Eye-tracking.** Eye tracking assessment was set up in a community health center annex in a room that was partitioned into separate spaces for the test participant and data collector using curtains [22]. Children were seated in front of a computer monitor (22-inch widescreen monitor, Dell Inc., TX) and a remote Pupil Centre Corneal Reflection (PCCR), 60-Hz eye-tracking camera (Tobii x2-60, Tobii technology, Stockholm, Sweden). A laptop computer was connected to the eye-tracker and a monitor to allow for data storing and stimulus presentation. A custom Python script, Psychopy functions [35] and a Tobii SDK plug-in were used to control stimulus presentation and data storing.

The data collector positioned the child so that the child's eyes were at an optimal distance and angle relative to the eye tracker (60 cm, ~0°). The data collector then calibrated an eye tracking camera for the participant's eyes and administered pro- and antisaccade tests. The saccade and antisaccade tasks were run in the same order for all children.

*Calibration.* A calibration target was an animation showing a yellow, smiling emoticon (initial size 4.7° x 2.6°). When the child looked at the target, it shrank to 2.8° x 1.6° and remained on the screen for 1000 ms, a period during which the eye tracker was calibrated. The outcome of a calibration procedure was evaluated visually against a predefined standard (i.e., a visualization of "good", "ok" and "bad" calibration outcomes) and the calibration was repeated up to two times if needed. The eye tracking system used in the study did not provide numeric data on the accuracy of the calibration.

*Prosaccade test.* In the pro-saccadic task, we assessed a latency of reflexive, visually triggered saccades. The child was instructed to maintain a stable position and look at the pictures on the screen. Each trial in the task started with a white fixation cross shown in the center of the screen. After the child looked at the fixation, it remained on the screen for 500 ms, followed by a saccade target on the left or right side of the screen. After the child had performed the prosaccade from the fixation to the target, the target remained on the screen for 1000 ms. The size (width x hight) of the saccade targets was 6.5° x 11.6°. The position of the target was chosen randomly for each trial with the constraint that the same total number of targets were presented on the left and right side of the screen. A minimum of 10 practice trials, followed by four sets of 16 test trials (64 test trials total) were administered for each child. During the practice trials, the saccade targets were yellow rectangular shapes with an instruction "Look!" written in the middle of the shape. If the child looked at the target, a smiley was superimposed on the rectangle as a reward. During the actual test, the targets were pictures of human faces against a variable background taken from the MUCT face database (www.milbo.org/muct). The size of the face pictures was standardized. No feedback during the test trials was given to the participants.

*Antisaccade test.* In the antisaccade task, the child was instructed to look at a fixation cross in the center, and when the picture on the left or right appeared, not to look at the picture, but to look away from it to the opposite direction. The child was instructed that if the picture appeared on the left, they were expected to make an eye movement to the right, and if the picture appeared to the right, to move their eyes to the left. Trials were started with a fixation stimulus presentation in the center of the screen, followed by a lateral stimulus on the left or right side of the screen. The task timing parameters were similar to those in the prosaccade task with the exception that the lateral stimulus remained on the screen for a total of 2 sec on each trial (giving the child a 2-sec period to make the correct antisaccade). The side of the lateral stimulus was chosen randomly with the exception that the same total number of stimuli was presented on the left and right side. The stimuli and their sizes and distances were the same as those in the saccade task. A minimum of 10 practice trials, followed by four sets of 16 test trials (64 test trials total) were administered for each child.

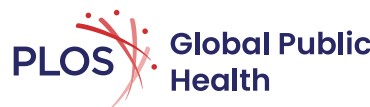

*Variables.* Prespecified inclusion criteria of successful trial were i) the starting position of the eyes was standardized so that the gaze was within the area of the central fixation stimulus (an 11.5° x 6.5° area in the center of the screen) before shifting to the area of the lateral target, ii) the duration of gaze shift from the central to the lateral stimulus was shorter than 50 ms (consistent with a direct path from the central to the lateral area), iii) the period from the start of the trial to the gaze entry to the target area did not have consecutive missing samples exceeding 100 ms, iv) a missing sample did not precede the gaze entry to the target area (i.e., the exact point of entry was known within the limits of the sampling frequency), and v) the SRT fell inside a time window that started 100 ms after target onset and ended 1000 ms after target onset. SRTs identified as outliers (2.5 SD from mean SRT) were excluded from statistical analyses (3.4% of all SRTs). Minimum of 10 valid test trials was required for the calculation of outcome variables for the final analysis.

The primary outcome in prosaccadic task was the mean of saccadic reaction times ($_p SRT_m$) (counted as sum of valid test prosaccadic times divided by number of valid tests), with $_p SRT_m$ defined as the time interval from the onset of the lateral target to the first entry of the gaze into the area of the lateral target (a 2.3° margin was added to the area of the target). Secondary outcome was the standard deviation of the saccadic reaction times ($_p SRT_{sd}$). In the antisaccadic task the primary outcome was the percentage of errors (PE), defined as eye movements towards the visual target (N error/N valid trials, where N valid trials is N errors+N correct saccades + N trials without a saccade (similar to Evdokinidis et al., 2002 [36]). Secondary outcomes in the antisaccade task were the mean latency of accurate eye movement ($LA_m$), defined as eye movements to the area opposite to that of the visual target, and standard deviation of the accurate eye movement ($LA_{sd}$), mean latency of error movement ($LE_m$) and standard deviation of error movement ($LE_{sd}$). Measures of coefficient of variation were obtained by diving the SD of reaction times by mean reaction time. (See S1 and S2 Figs for illustration of the tasks).

## Raven's coloured progressive matrices

Raven's coloured progressive matrices (CPM) measures non-verbal inductive reasoning and can be used also among illiterate people [37,38]. Test items become progressively more difficult (each series is more difficult than the previous), and cognitive ability is determined by the number of correctly completed matrices (0–36). Each correct answer provides a score of 1, each incorrect or no answer is counted as 0 and the test result is the sum of the answers. The reliability and criterion validity of CPM have been found to be good in many African countries and the test is believed to be a culturally and ethnically fair measurement tool [29,39].

Participants were asked to complete three series of twelve pictures that depicted 2 x 2 matrices of geometric shapes with one piece missing, and were asked to indicate the missing piece from an array of 6 alternatives by telling or pointing the correct answers to the data collector, who marked the answers to a score sheet. Two research assistants were trained to administer the Raven's test at the study clinic. Each test was administered by only one assistant.

## Child growth

The anthropometric measurements taken included the participant's height, weight, head circumference, and mid-upper arm circumference. All the anthropometric measurements were completed in triplicate. For the analysis, the mean of the first two readings was used if the reading did not differ by more than 0.1 kg for weight or 0.5 cm for other measurements. In case tolerance limit was exceeded, we calculated the mean from the pair of two measurements closest to each other. The 2006 WHO Child Growth Standard and the 2007 WHO Growth Reference for School-Aged Children and Adolescents were used for age-and-sex standardization of HAZ and height [40,41].

## Other collected data

Interview on socioeconomic status (SES) was performed. SES Z-score was created with principal component analysis by combining information on the building material of the house, main source of water, sanitary facility and ownership of

household items. This variable was used as a covariate in the analysis. Also, data of schooling was collected. There were many children who were not attending school at the time of the study but who had attended school earlier. In statistical analyses, we categorized these children to the group of no school years as we did not know the exact years of school attendance.

As per our statistical analysis plan, we used some maternal information as covariates (education of mother and intervention during pregnancy). This is a longitudinal follow-up study and some of the characteristics, like hemoglobin concentration of mother and literacy, can affect cognitive development of the offspring. This earlier collected information is provided in the Table 1.

## Statistical analyses

For this secondary analysis we used data from all the children who were still in the follow-up. We calculated mean eye-tracking and CPM results in the sample and assessed internal consistency of the measures by calculating correlations for the scores on the odd-numbered items with scores on the even-numbered items (i.e., split-half, odd-even correlation coefficients).

**Table 1. Baseline characteristics of the participating women at enrollment and their offspring at 13 years.**

| Maternal characteristic | N = 1320 |
|---|---|
| Age, years, mean (SD) | 24.9 (6.4) |
| Gestational age at enrollment, weeks, mean (SD) | 20.1 (3.1) |
| Gestational duration, weeks, mean (SD) | 38.5 (2.3) |
| Primiparous, n (%) | 306 (23.2) |
| HIV positive, n (%) | 161 (12.2) |
| Microsopic peripheral blood malaria parasitemia, n (%) | 117 (8.9) |
| Blood Hb concentration, g/L, mean (SD) | 110.2 (18.7) |
| Moderate or severe anemia, Hb < 100 g/L, n (%) | 351 (26.6) |
| Literate participants, n (%) | 384 (29.1) |
| Years of schooling completed, mean (SD) | 2.2 (2.7) |
| **Participant characteristics at 13 years** | |
| Anthropometrics, mean (n = 1002) | |
| Height (SD) | 142.8 (8.3) |
| HAZ (SD) | -1.7 (1.0) |
| Weight (SD) | 33.9 (6.6) |
| HC (SD) | 51.6 (1.5) |
| MUAC (SD) | 19.9 (2.1) |
| Literate participants, n (%) (n = 996) | 640 (64.3) |
| Blood Hb concentration, g/L, mean (SD) (n = 795) | 116 (14) |
| Years of schooling completed (n = 996) | |
| 0 years (%) | 156 (15.7) |
| 1–3 years (%) | 517 (51.9) |
| >3 years (%) | 323 (32.4) |
| Pubertal stage (n = 995) | |
| Stage 1 (%) | 492 (49.5) |
| Stage 2 (%) | 289 (29.1) |
| Stage 3–5 (%) | 214 (22.5) |

*Hb result from 795 participants at 13 years.

To assess whether the speed of visual orienting was associated with cognitive ability, we calculated Pearson correlation coefficient between $_pSRT_m$ and CPM score. Based on meta-analyses showing that the mean correlation between RT measures and fluid intelligence ranges from -.20 to -.26 in previous studies conducted in Western countries [42], we considered correlation coefficients > .20 as evidence for the hypothesized association. With a large sample size, low correlations can be significant, and the .20 cut-off was set as a criterion for a meaningful correlation.

To evaluate whether antisaccade PE add to the predictive value of the eye tracking tasks with respect to CPM, we constructed two regression models. In the first regression model, we entered speed ($_pSRT_m$) as a single predictor. In the second regression model, we added PE to the model. We compared the models using the likelihood ratio test. We considered statistically significant increase in prediction (at alpha < .05) as evidence in difference between the models and as evidence for the predictive value of PE. For the models adjusted for covariates, we used participants' age, sex, HAZ and head circumference at 13 years, schooling, maternal education at enrollment, the intervention during pregnancy and SES at 13 years of age [43,44].

To further evaluate the secondary outcomes, precisely $_pSRT_{sd}$ and 4 variables from the anti-saccadic task, we used the Pearson correlation coefficient among these variables. Correlations of the primary and secondary outcomes and CPM score were investigated. Again, we considered 0.20 cut-off separating meaningful correlations (at alpha < .05).

All analyses were performed using Stata version 15.1 or an equivalent statistical software package. The analysis plan is registered and can be found from https://osf.io/wndbp/files/osfstorage/639396cb084e0f01e2cdabd9.

The analysis plan was based on a previous study, which used least square regression models to examine the predictors of CPM in Malawian adolescents because this approach has proven to be robust against deviations from the normality assumption with a sample size that is smaller or similar to that of the current study [39]. Residual plots and Q-Q plots for the regression models are presented in S4 and S5 Figs 4. Sensitivity analyses testing the associations between eye tracking and CPM score by using non-parametric Spearman correlation tests are reported in S5 Table.

## Results

### Descriptive data

At approximately 13 years, we were able to collect data from 997 children on CPM, 760 on prosaccadic eye-tracking task, 618 on antisaccadic eye-tracking task and 1002 on anthropometrics (50,3% boys, Fig 1). There were no statistically significant differences in the background variables of the 13-year-old participants and the ones lost to follow-up except for maternal HIV prevalence, which was higher in the lost to follow up group (11% vs 22%, p < .001, S1 table). At baseline, the mothers were on average 24.9 years old and 23.2% were primiparous. Children were born at mean 38.5 gestational weeks. 29.1% of the mothers and 64.3% of children were literate. Mean years of schooling was 2.2 year for mothers and 2.8 for children. 15.7% of children had no schooling (Table 1).

Of the participants assessed with eye tracking, 760 had sufficient number of valid trials for the estimation of $_pSRT_m$ and 618 for the estimation of PE and the secondary eye tracking variables. $_pSRT_m$ and PE had moderate to high internal consistency (split-half, odd even Pearson r 0.83 and 0.98). CPM was successfully obtained from 997 (99.5%) participants and had moderate internal consistency of Pearson r 0.56 (Table 2). The distributions of $_pSRT_m$ and PE are presented in the S3 Fig.

### Association between eye tracking measures and CPM

$_pSRT_m$ was negatively correlated with CPM in line with our hypothesis (that is, faster $_pSRT_m$ was associated with higher CPM score), but the correlation was negligible (Pearson r = -0.12, Spearman correlation -0.08, p-values < .05). There was a very weak, negative correlation between PE and CPM (Pearson r = -0.11, Spearman correlation -0.08, p-values <0.05) (Fig 2).

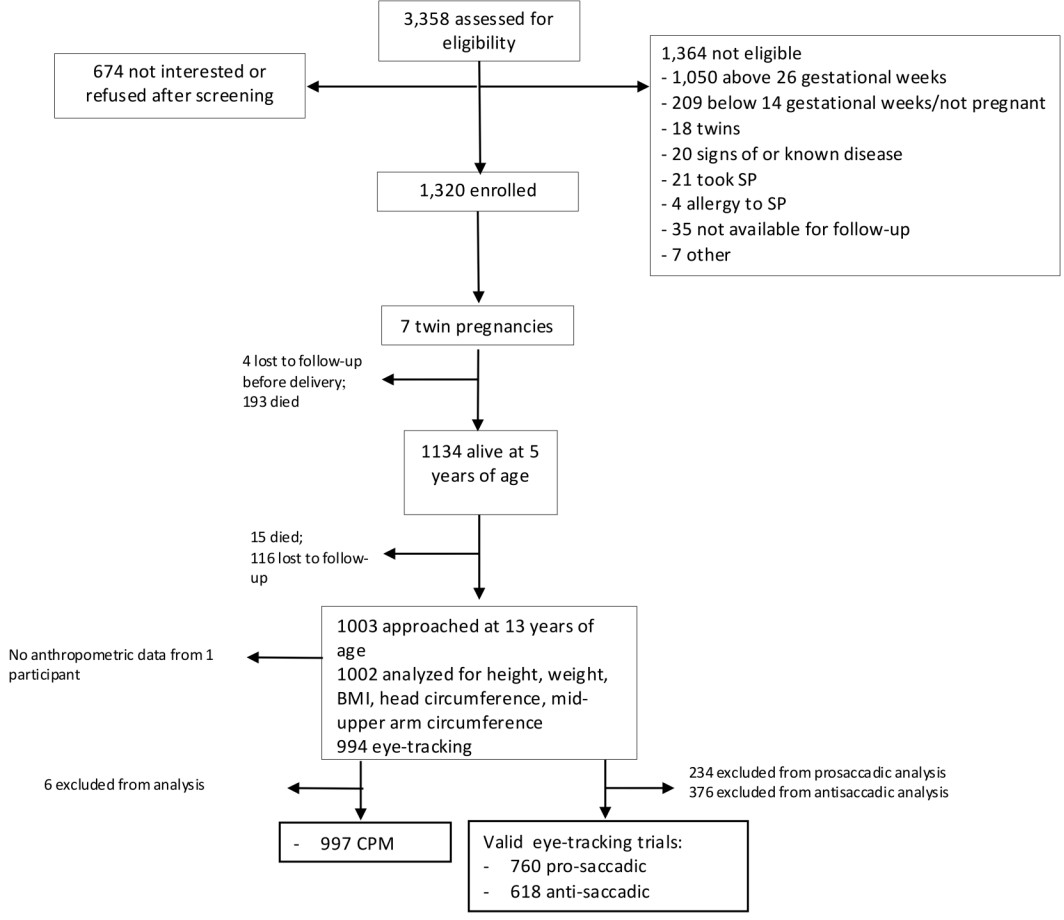

**Fig 1. Participant flow in CONSORT recommended format (Lancet 2001: 357: 1193).**

In regression analysis, $_pSRT_m$ result was significantly related to CPM score (coef -0.02, 95%CI -.04 to -.005, p = .01). Adding PE as a predictor improved model fit (LR test p = .004), with the model explaining 2% of variability in CPM scores (Table 3). The association between $_pSRT_m$ and CPM remained when the model was adjusted for co-variates whereas the association between PE and CPM was no longer significant (Table 3). Of the covariates, years of school completed predicted better performance in CPM (Model 1 coef. 0.56, 95%CI 0.39 to 0.73, Model 2 coef. 0.55, 95%CI 0.37 – 0.72, both p-values < .001), and female sex accounted lower scores in CPM (Model 1 coef. -0.99, 95%CI -1.63 to -0.35, p = .003, Model 2 coef. -0.96, 95%CI (-1.60 - -0.31), p = 0.004, S2 Table). Adjusted models explained 12% of the variance in CPM (Table 3).

The Pearson correlations of all eye-tracking variables (including secondary variables form the antisaccade task) and CPM score are presented in Table 4. None of the correlations with CPM score exceeded prespecified cut-off value (i.e., absolute correlation > .20). Mean latency of correct anti-saccades and its variation were correlated to mean and variation in error latency (correlations varying from 0.20 to 0.32, p-values < .001). All mean latency measures were positively correlated with the corresponding standard deviations, indicating that an increase in latency was accompanied by an increase in variability (correlations varying from 0.47 to 0.84, p-values < .001). The results of the sensitivity testing association between eye tracking variables with Spearman correlations were similar to the Pearson correlations (S5 Table).

**Table 2. Descriptive statistics for the Raven's coloured progressive matrices score and eye-tracking scores in the whole sample at 13 years of age.**

| | n | Mean (SD) | Valid trials (Mean, SD) | Split-half odd-even correlation (N) |
|---|---|---|---|---|
| Raven's score | 997 | 14.3 (3.8) | – | 0.56 (997) |
| Pro-saccadic: | | | | |
| $SRT_m$ (ms) | 760 | 213 (18.4) | 33.0 (12.0) | 0.83 (648) |
| $SRT_{sd}$ (ms) | 760 | 31.8 (9.7) | 33.0 (12.0) | 0.45 (648) |
| Anti-saccadic: | | | | |
| PE (%) | 618 | 51.6 (30.4) | 30.6 (12.3) | 0.98 (604) |
| $LA_m$ (ms) | 559 | 387 (105) | 30.6 (12.3) | 0.76 (151) |
| $LA_{sd}$ (ms) | 509 | 128 (59.2) | 30.6 (12.3) | 0.48 (151) |
| $LE_m$ (ms) | 618 | 260 (84.9) | 30.6 (12.3) | 0.40 (604) |
| $LE_{sd}$ (ms) | 604 | 100 (80.5) | 30.6 (12.3) | 0.32 (551) |

$SRT_m$, mean saccadic reaction time; $SRT_{sd}$, Standard deviation of saccadic reaction time; PE, percentage of errors in the antisaccade task; $LA_m$, mean latency of accurate antisaccades; $LA_{sd}$, standard deviation of accurate antisaccades; $LE_m$, mean latency of errors in the antisaccade task; $LE_{sd}$, standard deviation of error latency in the antisaccade task.

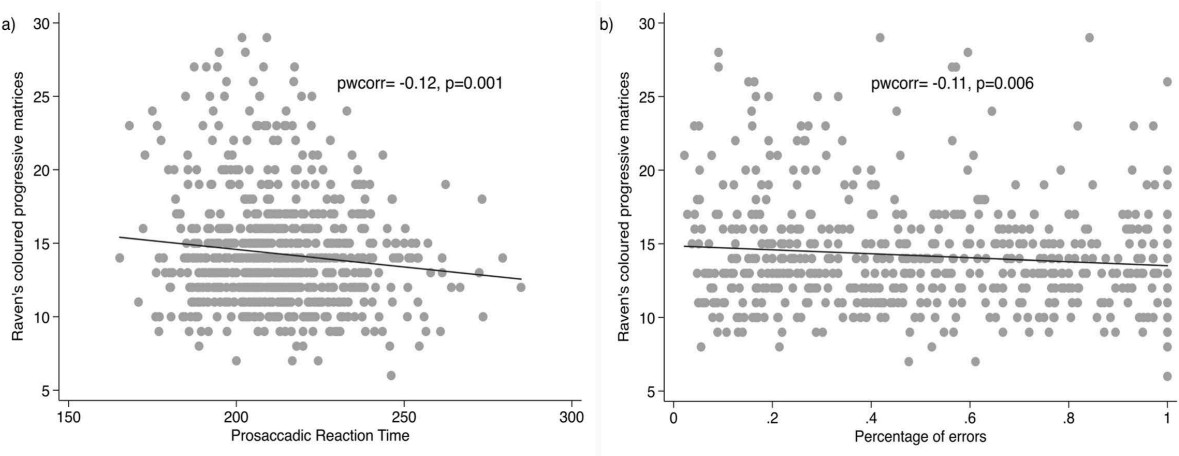

**Fig 2. Scatter plots of the correlations between a) prosaccadic mean reaction time and Raven's coloured progressive matrices score, and b) the percentage of errors and the Raven's score.**

## Post-hoc analyses

As a post-hoc analysis, we examined whether the association between eye tracking measures and cognitive ability was dependent on years in school. We fit a regression model, in which the eye tracking measures ($_pSRT_m$) and years in school were allowed to interact. The interaction between $_pSRT_m$ and years in school was significant (p < .006, S3 Table) as was the interaction between PE and years in school (p < .001, S4 Table). The associations between eye tracking measures and CPM were stronger amongst participants with increased number of completed years in school, as visualized in S4 Fig (median split of years in school was used in the visualization, but the original continuous variable was used in the regression models).

**Table 3. Association between eye-tracking results and Raven's coloured progressive matrices scores (CPM).**

**CPM score**

| Regressor | N | | Unadjusted model Coef. (95% CI) | P-value | Adjusted R-squared/ RMSE | LR test | N | Adjusted model[a] Coef (95% CI) | P-value | Adjusted R-squared/ RMSE | LR test |
|---|---|---|---|---|---|---|---|---|---|---|---|
| $SRT_m$ | 579 | | -0.02 (-0.04 - -0.005) | 0.01 | 0.01/ 3.7 | 0.004 | 567 | -0.02 (-0.03 - - 0.001) | 0.04 | 0.12/ 3.4 | 0.19 |
| $SRT_m$ and PE | 579 | $SRT_m$ | -0.02 (-0.04 - -0.006) | 0.008 | 0.02/ 3.7 | | 567 | -0.02 (-0.03- -0.002) | 0.03 | 0.12/ 3.4 | |
| | | PE | -1.47 (-2.46- -0.48) | 0.004 | | | | -0.63 (-1.60 - 0.33) | 0.20 | | |

$SRT_m$, Mean prosaccadic reaction time; PE, percentage of errors.

[a]adjusted for participant age, sex, HAZ at 13 years, head circumference, schooling, and maternal education, the intervention during pregnancy and socioeconomic status at 13 years.

Analysis is done including the maximal amount of the participants (with available the data required for the testing).

**Table 4. Pearson correlation coefficients among 7 different eye-tracking tasks and Raven's coloured progressive matrices score (CPM) at 13 years of age.**

| $SRT_{sd}$ | PE | $LA_m$ | $LA_{sd}$ | $LE_m$ | $LE_{sd}$ | | CPM |
|---|---|---|---|---|---|---|---|
| 0.61** | -0.04 | 0.11* | 0.06 | 0.23** | 0.12* | $SRT_m$ | -0.12* |
| | -0.07 | 0.02 | 0.08 | 0.14** | 0.08 | $SRT_{sd}$ | -0.03 |
| | | -0.13* | -0.10* | -0.19** | -0.19** | PE | -0.11* |
| | | | 0.47** | 0.32** | 0.27** | $LA_m$ | -0.06 |
| | | | | 0.20** | 0.20** | $LA_{sd}$ | -0.01 |
| | | | | | 0.84** | $LE_m$ | -0.05 |
| | | | | | | $LE_{sd}$ | -0.00 |

$SRT_m$, Prosaccadic reaction time, mean; $SRT_{sd}$, Reaction time, standard deviation; PE, percentage of errors; $LA_m$, mean latency of accurate eye movement; $LA_{sd}$, standard deviation of accurate eye movement; $LE_m$, mean latency of error movement; $LE_{sd}$, standard deviationof error eye movement; CPM, Raven's coloured progressive matrices.

N varies from 489 to 757, all the participants with data from each measurement included in the analysis.

*p<0.05.

**p<0.001.

## Discussion

Our aim was to evaluate how eye tracking measures of speed and control are associated with CPM – a well-established test of non-verbal reasoning and a marker of the general factor of intelligence, in rural Malawian adolescent population [45]. There was a weak and statistically significant correlation between eye-tracking measured mean prosaccadic reaction time ($_pSRT_m$) and CPM in the predicted direction, although the correlation was smaller than expected not exceeding the set 0.20 cut-off. This correlation remained significant after adjusting for covariates, including schooling. A similar, very weak association was also observed between PE and CPM score, but this association was not significant in the adjusted analysis. Thus, our results give very weak support for the hypothesized correlation between eye-tracking measures and CPM score.

There are limitations that might have weakened the expected associations between eye tracking measures and CPM. Measurement error may lead to a significant underestimation of the size of the true correlation between the measures [46]. For this reason, it is important to consider different sources of measurement error in our main outcomes (i.e., $_pSRT_m$, PE, and CPM) and the potential contribution of this e on the examined associations. In general, measurement error may arise from variations in test conditions, measurement instruments, missing data, and measurement bias. For example,

light circumstances, instrumental imprecision, experimenter-dependent causes, and in addition, test subject's fatigue, mood, hours of sleep the previous night, etc. can cause noise to the results. Many of these factors are difficult to perfectly control among human participants and experimenters. Measurement error is typically estimated by calculating test-retest reliability estimates across repeated testings and controlled for by calculating estimates of a latent construct [47], but such data were not available in our study.

However, all children in the current study were assessed in the same room, with the same equipment, and by the same experienced assessors. With standardized operating procedures, practice trials prior to actual testing and quality forms used, we tried to assure standardized test settings and test performance for each participant. Although the low sampling rate (60 Hz) of our eye tracking system add to the imprecision of the measurements, the split-half/odd-even correlations still showed moderate to good internal consistency for the key tests used. These considerations suggest that our measures were not affected by unusual amounts of instrumental noise.

Speed of information processing is believed to be a fundamental aspect of brain function and a good marker of individual difference in cognitive capacity [47]. Neural connectivity, especially parieto-frontal integration [48], and myelination [49], for instance, may partly explain the variations in processing speed. There are variety of factors that can affect brain development and myelination among children living in low-resource settings, such as early inflammation, nutrition, nurturing care and home environment, to mention some [50]. Aside from these factors, the speed of processing may improve with practice [51]. It is, therefore, possible that variation in prosaccadic reaction time captures the effects of multiple factors in the child's growth environment that are relevant for cognitive development. However, our results show that prosaccadic reaction time alone is not sufficiently strongly associated with cognition to have clear practical utility in low resource setting.

The antisaccadic task directional error (PE) decreases with age due to pruning and practicing [14,36,52]. The performance of the antisaccadic task is regulated by many cortical and subcortical brain areas, and the ability to suppress involuntary responses is crucial for daily living [14]. In addition, working memory and attention has been shown to predict antisaccadic performance [53]. In earlier studies, PE has varied from 0 and 30% in adult population, and 40–50% in children [11,15,36]. High PE is typical in the beginning of the task, and as a result of learning, there are less errors. Enduring high levels of errors have been related to ADHD [13], frontal lobe lesions [54] or mental and neurological problems [55].

One plausible explanation of the high PE in our sample could be the age of the participants [15]. Similar result of high PE (mean 56%) was found in a study in Blantyre, Malawi among children suffering from cerebral malaria and controls, and was interpreted to reflect the adverse effects of the environment [52]. A negative correlation (-0.22) between PE and CPM has been previously found in a study with Greek young adults [36] which is in line with our statistically significant, but negligible finding. Compared to the prosaccade task, the antisaccade task has closer links to executive control and working memory [56], and is believed to be regulated by partly different parts of the brain (i.e., dorsolateral prefrontal cortex, anterior cingulate cortex) [57]. Our regression models showed that after controlling for covariates, prosaccadic reaction time still weakly predicts CPM, but PE does not. This finding suggests that antisaccadic test does not contribute independent predictive value for assessing cognitive development in LMIC setting among adolescent population.

Lastly, it is important to consider the success of administering CPM in the current population. In this study CPM test was used as a measurement method of fluid, non-verbal intelligence [58]. This test has been developed in 1938 and revised twice [37]. Internal consistency and split-half reliabilities of the test were found to be good in Australian child sample, with Spearman-Brown corrected split half reliabilities ranging from.81 to.90 for 6–11-year-old children [38]. In the current study, the Spearman-Brown corrected split half reliability (.72) and the mean test scores (14.3) were lower than those in the Australian sample, but the mean scores were comparable to a previous study among Malawian 15-year-old participants (Mean = 15, SD 6, range 0–33). In this Malawian study, female sex was associated with poorer performance in CPM, similar to our findings [39]. In a study performed in Ghana among 185 children aged 6–12 years, age and SES (estimated by the attendance of private or public school) were positively associated CPM [58]. Thus, the results showing moderate

split-half reliability for the CPM in the current study as well as the observed association with schooling, give some support for the reliability and criterion validity of CPM as a proxy measure of cognitive ability in the current study context [29,59]. However, several factors, such as fatigue [60] may contribute to CPM performance, and the use of Raven's test as a measure of general intelligence in Sub-Saharan Africa has also been questioned [29]. These factors limit the interpretation of the results.

Our post hoc analyses suggested that eye tracking measures may be more predictive of cognitive outcomes in the subgroup of children who had to some or all years of compulsory primary schooling. However, we note that this result was based on an unplanned exploratory analysis. As such, it should be regarded as a potentially interesting tentative result and should be subjected to further confirmatory analyses in new datasets.

## Conclusion

Our study suggests that processing speed, measured as prosaccadic reaction time, is a very weak correlate of cognitive ability in children living in rural Malawi. The eye tracking measure of response control, in turn, was not correlated with cognitive ability after controlling for schooling. We cannot rule out the possibility that these correlations are partially explained by measurement error, which could be minimized by repeated testing. Our results do suggest, however, that even though $_pSRT_m$ and CPM are associated in a theoretically meaningful way, the predictive value of single measurements of $_pSRT$ or PE, as currently implemented, is not strong enough to have practical or clinical utility - or generalizability according to these results alone - as a sole marker of cognitive ability development in a low-resource setting.

## Supporting information

**S1 Fig. Prosaccade reaction time ($_pSRT$) task and data. a) Illustration of a single trial in the $_pSRT$ task.** After looking towards the fixation stimulus in the center of the screen, a lateral target stimulus was shown on the left or right side of the screen. The lateral target was a picture of a face in a rectangular frame. The participant was instructed to look at the lateral target as quickly as possible. Recorded xy-coordinates of gaze are shown and numbered from 1 to n samples. b) Gaze traces showing the x- and y-coordinates of gaze as a function of time for a single trial. $_pSRT$ was defined as the point at which the gaze shifted from the center of the screen to the lateral target. The x-coordinate of the borders of the central and lateral areas are shown by grey and green dashed lines. Median-filtered gaze samples are shown with dots and solid lines, raw samples as dashed lines. c) Gaze traces for all valid trials for the example observer.
(DOCX)

**S2 Fig. Antisaccade reaction time (aSRT) task and data. a-b) Gaze samples on a single trial with a correct aSRT. c) Gaze samples for all valid aSRTs for one example observer, including correct aSRTs as well as trial on which there was no saccade (i.e., gaze did not leave central AOI). d-e) Gaze samples from trials with a prosaccade error.** Prosaccade error (PE) rate was estimated as a proportion of trials with a PE out of all valid trials (i.e., sum of trials with a correct aSRT, trials on which the gaze did not leave the center AOI and trials with a PE).
(DOCX)

**S3 Fig. Distribution of a) mean prosaccadic rection time ($_pSRT_m$) and b) percentage of errors (PE).**
(DOCX)

**S4 Fig. The assocation between eye tracking measures and Ravens coloured progressive matrices (CPM) score by years in school.** Scatterplots of prosaccadic reaction time ($_pSRT_m$) and CPM score by a median split of years in school on the left. Scatterplots of PE and CPM score by median split of years in school on the right. Median split of years in school was used in the visualization, but the original continuous variable was used in the regression models.
(DOCX)

**S5 Fig.** a-b) Residual plots of the unadjusted models: a) model where Raven's coloured progressive matrices (CPM) is dependent variable and prosaccadic reaction time (SRT) independent variable, b) model where CPM is dependent variable, SRT and percentage of errors (PE) independent variables. c-d) Corresponding plots for adjusted models: c) model where CPM is dependent variable and SRT independent variable, d) model where CPM is dependent variable, SRT and PE independent variables. Models were adjusted for participant age, sex, height-for-age Z-score at 13 years, head circumference, schooling, and maternal education, the intervention during pregnancy, and socioeconomic status at 13 years. (DOCX)

**S6 Fig.** a-b) QQ plots of the unadjusted models: a) model where Raven's coloured progressive matrices (CPM) is dependent variable and prosaccadic reaction time (SRT) independent variable, b) model where CPM is dependent variable, SRT and percentage of errors (PE) independent variables. c-d) Corresponding plots for adjusted models: c) model where CPM is dependent variable and SRT independent variable, d) model where CPM is dependent variable, SRT and PE independent variables. Models were adjusted for participant age, sex, height-for-age Z-score at 13 years, head circumference, schooling, and maternal education, the intervention during pregnancy, and socioeconomic status at 13 years. (DOCX)

**S1 Table. Baseline characteristics of the mothers of the included (follow-up) and excluded (lost to follow-up) participants at approximately 13 years of age.**
(DOCX)

**S2 Table. The association between eye-tracking results and Raven's coloured progressive matrices score (CPM), unadjusted and adjusted models.**
(DOCX)

**S3 Table. Summary of a regression model with prosaccadic reaction time ($_p srt_m$), years of school completed, and the interaction of $_p srt_m$ and years of school completed as predictors of Raven's coloured progressive matrices score (CPM) score.**
(DOCX)

**S4 Table. Summary of a regression model with percentage errors (PE), years of school completed, and the interaction of PE and years of school completed as predictors of Raven's coloured progressive matrices score (CPM) score.**
(DOCX)

**S5 Table. Spearman correlation coefficients among 7 different eye-tracking tasks and Raven's coloured progressive matrices score (CPM) at 13 years of age.**
(DOCX)

## Ackowledgements
We thank the study participants and the study staff.

## Author contributions
**Conceptualization:** Ulla Ashorn, Per Ashorn, Jukka M. Leppänen.

**Data curation:** Karoliina Videman.

**Formal analysis:** Karoliina Videman, Lotta Hallamaa, Jukka M. Leppänen.

**Funding acquisition:** Ulla Ashorn, Per Ashorn.

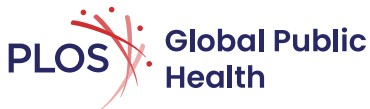

**Investigation:** Ulla Ashorn, Charles Mangani.

**Methodology:** Ulla Ashorn, Per Ashorn, Charles Mangani, Jukka M. Leppänen.

**Project administration:** Ulla Ashorn, Charles Mangani.

**Supervision:** Ulla Ashorn, Per Ashorn, Jukka M. Leppänen.

**Writing – original draft:** Karoliina Videman.

**Writing – review & editing:** Karoliina Videman, Ulla Ashorn, Per Ashorn, Lotta Hallamaa, Kenneth Maleta, Charles Mangani, Jukka M. Leppänen.

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
