## [Decision Letter · Decision Letter 0]

PGPH-D-24-01531

Eye-tracking measures of oculomotor speed and control as markers of cognitive ability in Malawian adolescent population: Secondary analysis of a randomized controlled trial

Dear Dr. Videman,

Thank you for submitting your manuscript to PLOS Global Public Health. After careful consideration, we feel that it has merit but does not fully meet PLOS Global Public Health’s publication criteria as it currently stands. Therefore, we invite you to submit a revised version of the manuscript that addresses the points raised during the review process.

A rebuttal letter that responds to each point raised by the editor and reviewer(s). You should upload this letter as a separate file labeled 'Response to Reviewers'.A marked-up copy of your manuscript that highlights changes made to the original version. You should upload this as a separate file labeled 'Revised Manuscript with Track Changes'.An unmarked version of your revised paper without tracked changes. You should upload this as a separate file labeled 'Manuscript'

We look forward to receiving your revised manuscript.

Kind regards,

Valentina Gallo

Academic Editor

Journal Requirements:

1. Please include a complete copy of PLOS’ questionnaire on inclusivity in global research in your revised manuscript. Our policy for research in this area aims to improve transparency in the reporting of research performed outside of researchers’ own country or community. The policy applies to researchers who have travelled to a different country to conduct research, research with Indigenous populations or their lands, and research on cultural artefacts. The questionnaire can also be requested at the journal’s discretion for any other submissions, even if these conditions are not met.  Please find more information on the policy and a link to download a blank copy of the questionnaire here: https://journals.plos.org/plosone/s/best-practices-in-research-reporting. Please upload a completed version of your questionnaire as Supporting Information when you resubmit your manuscript.

2. Your current Financial Disclosure states is different from your funding information on the submission form. Please indicate by return email the full and correct funding information for your study and confirm the order in which funding contributions should appear. Please be sure to indicate whether the funders played any role in the study design, data collection and analysis, decision to publish, or preparation of the manuscript.

3. In the online submission form, you indicated that "Data Availability Statement: Data are available upon request due to concerns about participant confidentiality. Data requests from researchers who meet the criteria for access to confidential information can be made to Dr. Charles Mangani (cmangani@kuhes.ac.mw).". 

3. Uploaded as supplementary information.

4. We noticed that you used “data not shown” in the manuscript. We do not allow these references, as the PLOS data access policy requires that all data be either published with the manuscript or made available in a publicly accessible database. Please amend the supplementary material to include the referenced data or remove the references.

5. Please provide an Author Summary. This should appear in your manuscript between the Abstract (if applicable) and the Introduction, and should be 150–200 words long. The aim should be to make your findings accessible to a wide audience that includes both scientists and non-scientists. Sample summaries can be found on our website under Submission Guidelines: 

https://journals.plos.org/globalpublichealth/s/submission-guidelines#loc-parts-of-a-submission

6. We have noticed that you have uploaded Supporting Information files, but you have not included a list of legends. Please add a full list of legends for your Supporting Information files after the references list. 

Additional Editor Comments (if provided):

Reviewers' comments:

Reviewer's Responses to Questions

**Comments to the Author**

1. Does this manuscript meet PLOS Global Public Health’s publication criteria ? Is the manuscript technically sound, and do the data support the conclusions? The manuscript must describe methodologically and ethically rigorous research with conclusions that are appropriately drawn based on the data presented.

Reviewer #1: Yes

Reviewer #2: Partly

2. Has the statistical analysis been performed appropriately and rigorously?

Reviewer #1: No

Reviewer #2: Yes

3. Have the authors made all data underlying the findings in their manuscript fully available (please refer to the Data Availability Statement at the start of the manuscript PDF file)?

Reviewer #1: Yes

Reviewer #2: Yes

4. Is the manuscript presented in an intelligible fashion and written in standard English?

Reviewer #1: Yes

Reviewer #2: Yes

5. Review Comments to the Author

Reviewer #1: This is a secondary analysis of outcomes from a large Malawian adolescent study on ocular speed and cognition. The good news is that it appears to be a rich data set and power is not an issue with such large sample sizes. The authors also present reasonable specific hypotheses. Correlations and regression analyses were run. I agree with the decision to use 0.20 as a cutoff, but would have preferred Spearman to Pearson because of distributional assumptions. While the analysis seems appropriate, there are no regression diagnostics reported, such as residual plots, Q-Q plots, and other items that any introductory regression class would require of it's student projects. I don't know if distributional assumptions are appropriate or if the model accounts for heterogeneity. Principal components were used to create a cognition composite, but no eigenvalue loadings are given. What is the final composite? In general, this is an interesting study with unreported information that the statistician probably has but did not report.

Reviewer #2: TITLE: Eye-tracking measures of oculomotor speed and control as markers of cognitive ability in Malawian adolescent population: Secondary analysis of a randomized controlled trial

COMMENT

Authors should please check their study goal/aim. In line 87 – 89, “…was to further examine the informational value and utility of eye-tracking tests of processing speed and control as measures of cognitive development in low-resource settings”. Then, at the beginning of the discussion section, “…Our aim was to evaluate how eye tracking measures of speed and control are associated to CPM”

Line 90 -91 “Our a priori hypotheses were that faster sensory-motor speed, as indicated by mean pro-saccadic…”. Please revise for clarity

Please justify the inclusion of ‘maternal characteristics’ in Table 1, as their relevance to the current study is unclear

Line 142- 143 “The study participants were approximately 13-year-old at the time of the follow-up”.

Please clarify how the participants were approximately 13 years old at the time of the follow-up, given that the original study was conducted between 2003 and 2006 among pregnant women

Regarding Discussion Section:

The discussion section requires further clarification and elaboration to strengthen its overall coherence and impact

The positioning of ‘7 twin pregnancies’ in the figure 7 is unclear and should be revised for better clarity and visual understanding.

6. PLOS authors have the option to publish the peer review history of their article (what does this mean? ). If published, this will include your full peer review and any attached files.

**Do you want your identity to be public for this peer review?** For information about this choice, including consent withdrawal, please see our Privacy Policy .

Reviewer #1: No

Reviewer #2: No

---

## [Decision Letter · Decision Letter 1]

Eye-tracking measures of oculomotor speed and control as markers of cognitive ability in Malawian adolescent population: Secondary analysis of a randomized controlled trial

PGPH-D-24-01531R1

Dear MD Videman,

We are pleased to inform you that your manuscript 'Eye-tracking measures of oculomotor speed and control as markers of cognitive ability in Malawian adolescent population: Secondary analysis of a randomized controlled trial' has been provisionally accepted for publication in PLOS Global Public Health.

Best regards,

Julia Robinson

Executive Editor

Reviewer Comments (if any, and for reference):

Reviewer's Responses to Questions

**Comments to the Author**

1. If the authors have adequately addressed your comments raised in a previous round of review and you feel that this manuscript is now acceptable for publication, you may indicate that here to bypass the “Comments to the Author” section, enter your conflict of interest statement in the “Confidential to Editor” section, and submit your "Accept" recommendation.

Reviewer #1: All comments have been addressed

2. Does this manuscript meet PLOS Global Public Health’s publication criteria ? Is the manuscript technically sound, and do the data support the conclusions? The manuscript must describe methodologically and ethically rigorous research with conclusions that are appropriately drawn based on the data presented.

Reviewer #1: (No Response)

3. Has the statistical analysis been performed appropriately and rigorously?

Reviewer #1: (No Response)

4. Have the authors made all data underlying the findings in their manuscript fully available (please refer to the Data Availability Statement at the start of the manuscript PDF file)?

Reviewer #1: (No Response)

5. Is the manuscript presented in an intelligible fashion and written in standard English?

Reviewer #1: (No Response)

6. Review Comments to the Author

Reviewer #1: (No Response)

7. PLOS authors have the option to publish the peer review history of their article (what does this mean? ). If published, this will include your full peer review and any attached files.

**Do you want your identity to be public for this peer review?** For information about this choice, including consent withdrawal, please see our Privacy Policy .

Reviewer #1: No
